# Impact of Chemical Insecticide Application on Beneficial Insects in Maize

**DOI:** 10.3390/insects15040224

**Published:** 2024-03-25

**Authors:** Giuseppe Camerini, Stefano Maini, Lidia Limonta

**Affiliations:** 1Istituto di Istruzione Superiore Taramelli-Foscolo, Via Mascheroni, 53, 27100 Pavia, Italy; giuseppe.camerini@taramellifoscolo.it; 2Dipartimento di Scienze e Tecnologie Agro-Alimentari-Entomologia, Università di Bologna, Viale Fanin, 42, 40127 Bologna, Italy; stefano.maini@unibo.it; 3Dipartimento di Scienze per gli Alimenti, la Nutrizione e l’Ambiente, Università “degli Studi” di Milano, Via Celoria 2, 20133 Milano, Italy

**Keywords:** *Trichogramma brassicae*, aphids, predators, parasitoids, agricultural maize practices, insecticide side effects, northern Italy

## Abstract

**Simple Summary:**

In the Po Valley, the control of maize from insect pests (*Ostrinia nubilalis* and *Diabrotica virgifera virgifera*) can include insecticide applications. The results of research on insecticide spray’s impact on beneficial insects, which can contribute to the biological control of maize pest, are reported. Aphid predators and *Trichogramma brassicae* (*O. nubilalis* egg parasitoid) were used as indicators. Chemical insecticide application significantly increased aphid abundance when no rotation protocol was applied. In addition, an alteration in aphid predator community composition was recorded. In insecticide-sprayed fields managed according to repeated crop protocols, the predator community was dominated by hoverflies, while in maize fields where crop rotation was practiced, ladybirds and *Orius* spp. predominated. In addition, the insecticides had a negative effect on the natural parasitism of *T. brassicae*. The research suggests that Integrated Pest Management strategies should be planned based on crop rotation protocols and biological control of maize pests.

**Abstract:**

The European corn borer (ECB) (*Ostrinia nubilalis* Hübner) and to a lesser extent the western corn rootworm (*Diabrotica virgifera virgifera* LeConte) are a threat to maize in the Po Valley (Northern Italy), and their control can require insecticide applications. The results of a study to evaluate the effects of insecticide sprays on the beneficial insect *Trichogramma brassicae* (Hymenoptera: Trichogrammatidae) and aphid predators are reported. A three-year research project was carried out in two Study Areas, in Lombardy. In area 1, crop rotation was a common practice, while in area 2 repeated maize crop was practiced. The natural trend of ECB egg masses attacked by *T. brassicae* was affected and parasitism rates were reduced as a result of insecticide exposure (chlorpyriphos methyl, cypermethrin, alphacypermethrine). Repeated maize crop and insecticides spraying increased the abundance of the aphid population and negatively affected the aphid predator community, which mainly included ladybirds, hoverflies, true bugs and lacewings. The predator community was dominated by hoverflies in sprayed fields managed according to repeated maize crop protocols, whereas ladybirds and *Orius* spp. dominated in maize fields managed according to crop rotation protocols. Crop rotation protocols help to prevent ECB outbreaks; when the risk of exceeding the economic threshold limit is high, and this may be the case when maize is cultivated for seeds or for horticultural crops such as sweet corn, inundative release of *T. brassicae* and/or microbial control (i.e., use of *Bacillus thuringiensis* preparations) can integrate natural biocontrol, and provide a valuable alternative to chemical insecticides.

## 1. Introduction

Maize (*Zea mays* L.) is a widespread cereal crop throughout the World, including Italy. The majority of Italian maize is cultivated in the Po floodplain (Northern Italy) [1]. The European Corn Borer (ECB—*Ostrinia nubilalis* Hübner, Lepidoptera: Crambidae) is historically the main entomological pest to maize in this area [2,3,4]. More recently, the Po Valley, like many other European agricultural districts, was invaded by a new exotic maize pest, the Western Corn Rootworm (WCR—*Diabrotica virgifera virgifera* LeConte, Coleoptera: Chrysomelidae) [5,6]. Yield losses caused by WCR are mainly due to larvae feeding on roots. With the exception of spiders, which can prey on WCR adults, the effectiveness of natural enemies is poor [5].

Damage caused by these two pests can be conveniently prevented by planning appropriate crop rotation protocols, a precaution that helps to keep the pests below the economic threshold limit using traps to monitor and study adult flight and movements [7,8,9,10,11,12,13]. Both WCR and ECB are in general the main pests of maize; crop rotation is recommended to limit these two pests [12,13]. Moreover, post-harvest stubble shredding and avoiding sod seedling or no-tillage practices help limit WCR eggs and ECB overwintering mature larvae. Later on, with the growth of maize plants, the damage caused by ECB larvae can be significant; meanwhile, WCR adults start feeding on upper leaves. These two main pests develop especially in those agricultural districts where maize is the dominant cereal species, and area-wide crop and repeated maize crop are common practices [3,6,13]. In the Po Valley, WCR has one generation per year and over winter as eggs [6]. ECB usually develops two or three generations: adults coming from the overwintering larvae start flying in May [14]. First-generation larvae pupate in July, building up to second-flight moths that will lay eggs. The second-generation larvae are responsible for more severe damage, i.e., attacks on ear shank that can cause cob drops during mechanical harvesting [1]. Moreover, in sweet maize the presence of larvae in kernels negatively affects the quality of the product [2]. Probably due to the climate change, a third adult flight in mid-September and third-generation larvae, which normally do not reach maturity, have been observed in recent years [4].

In the Po Valley, maize farms rely mainly on crop rotation and wild natural enemies to limit ECB populations. *Trichogramma brassicae* Bezdenko (Hymenoptera: Trichogrammatidae) and the larval parasitoid *Lydella thompsoni* Herting (Diptera: Tachinidae) are useful to support such a strategy. However, where the threshold limit is particularly low (i.e., sweet maize and maize for seed), ECB is limited by insecticide sprays or inundative *T. brassicae* releases [2]. Natural control by *Trichogramma* egg parasitoids is active against ECB throughout all the periods of overlapping generations of the host [11,12,13,15,16,17].

Aphids (Hemiptera: Aphididae), in the Po Valley, are considered as maize secondary pests. Their natural enemies are mainly ladybirds (Coleoptera: Coccinellidae), hoverflies (Diptera: Syrphidae) predatory bugs (Hemiptera: Anthocoridae) and lacewings (Neuroptera: Chrysopidae) [1]. Normally, these predators are able to naturally control aphid colonies in the maize agroecosystems of northern Italy (authors personal observation).

Our aim was to study the effects of insecticide treatments on ECB and WCR in two agricultural districts using different maize cultivation techniques (repeated maize crop vs. crop rotation). Here, we report the results of comparative observations carried out in maize fields that were located in farms which adopted chemical control protocols against WCR and ECB, compared to untreated fields. Side effects of insecticides on non-target species were assessed. The aphid predators and *T. brassicae* were chosen to indicate acute toxicity induced by chemical treatments.

## 2. Materials and Methods

### 2.1. Study Areas

Two areas were studied. In both agricultural districts the maize is traditionally sown in March: maize hybrids usually sown belong to FAO classes 400, 500 and 600. The first Study Area (Study Area 1) was located at 45°05′23″ N 9°04′58″ E in the municipality of Bastida Pancarana (Pavia Province, central western Po flood plain). The landscape is dominated by arable land (cereals) interspersed with legumes and alfalfa. Maize is grown for the production of feed grain or sweet maize. Maize is not usually supported by irrigation and the majority of farms are managed according to crop rotation protocols. Insecticide use in maize is not the rule. Both sprayed and unsprayed fields were studied in the 2008 and 2009 seasons to assess the impact of insecticides on aphid predators and *T. brassicae*. In addition, aphid predators were surveyed in an unsprayed maize field in 2010. In this Study Area, water pan traps (n = 16), baited with ECB sex pheromone (bait = 3:97 Z:E-14 tda rubber stopper 0.1 mg, renewed every 14 days) and phenylacetaldehyde (370 mg on 25 × 25 mm filter paper 2.7 mm thick) were installed and regularly checked (seasons 2008–2010) in order to monitor the diel flight rhythms of both ECB adult males and females [7]. The size of fields used for sampling ranged from 1.5 ha to 6.8 ha.

The second Study Area (Study Area 2) is located in the central Po floodplain (Santo Stefano Ticino 45°29′02″ N 8°53′57″ E—Robecco sul Naviglio—45°26′50″ N 8°52′42″ E). The degree of urbanization is much higher than in Study Area 1. Another difference concerns the final destination of the crop; maize biomass is used entirely for silage. The maize fields used for the trials belonged to a farm that did not apply seasonal crop rotation. A network of canals provided plenty of water for crop irrigation. It is important to note that the maize in the Robecco sul Naviglio area was not subjected to chemical spraying throughout the research period (2008–2010) and was therefore considered as a control sample in 2008 and 2009. On the other hand, Santo Stefano Ticino was sprayed in 2008 and 2009, while no insecticide was applied in 2010. The size of fields used for sampling ranged from 1.9 ha to 6.1 ha.

### 2.2. Fieldwork Methods

#### 2.2.1. Aphids

Aphids and their predators were studied in Bastida Pancarana (Study Area 1) and in Santo Stefano Ticino and Robecco sul Naviglio (Study Area 2). The aim of the first research season (2008) was to analyze the composition, phenology and distribution of aphid communities on maize plant organs. The Italian countryside is generally made up of small fields. The size of the edges can affect the distribution of pests and natural enemies, and maize cultivation is no exception [18]. For this reason, samples included groups of maize plants located both in the inner sector of the fields and along edges. Rectangular plots were used as study sites and this choice made it easier to place samples. Edge and inner samples were made visible by attaching a beam to a maize plant with a red ribbon tied to it. The edge samples were located in the middle of the minor sides, at a distance of 8 m from the field edge. In 2008, the presence/absence of aphids on stalk, ears and on ear axil leaves was assessed.

In both Study Areas, a sample of 160 maize plants was examined; such a sample comprised 4 subsamples of 40 plants; and 2 subsamples were considered at opposite edges of the field. Another 2 subsamples were placed across the inner section of the field. Subsamples included 4 replicates consisting of 10 plants in a row.

In 2009 and 2010, a more informative survey was carried out: aphid abundance on each maize plant was measured according to 4 abundance classes: no presence, 1–10 specimens, 11–100 specimens, and >100 specimens. Ladybirds (larvae, pupae, adults), hoverflies (larvae, pupae), true bugs (nymphs and adults) and other predators of minor importance found on aphid-infested plants were also surveyed and identified. A sample of hoverfly pupae was collected and transferred to the lab for rearing.

In both sprayed and unsprayed sampling areas, aphid and predator abundance was assessed once, 4 weeks after insecticide application. Chemical sprays were targeted to control ECB larvae and WCR adults. Two fields per year were used for sampling in each of the study areas, except for study area 1 in 2010, when one field was used for sampling.

In 2008 and 2009 a chlorpyriphos methyl—cypermethrin mixture (“Daskor”, 2 kg/ha) was sprayed in Study Area 1. In Study Area 2 the insecticides “Contest” (0.4 kg/ha—active ingredient: alpha-Cypermethrin) and “Cresit” (0.55 kg/ha—Teflubenzuron) were applied on 21 July while in 2009 “Contest” (0.5 kg/ha—active ingredient: alpha-Cypermethrin) was applied on 11 July in Study Area 2.

#### 2.2.2. ECB Egg Masses Parasitism by *Trichogramma brassicae*

ECB egg masses sampled for analysis of parasitism by *T. brassicae* consisted of two subsamples of at least 10 egg masses each. One subsample was collected from the edge of the maize field and the other from the center of the field. The criteria used to position and mark the sampling sectors of the maize fields were the same as those used for the aphid survey. Maize plants around the poles marking the edge and inner points were randomly surveyed for ECB eggs. Egg masses were sampled in both sprayed and untreated fields in Study Area 1. Eggs were detected by carefully examining the ears and leaves of the maize plants. ECB egg masses found in maize fields were transferred to a plastic Petri dish (35 mm diameter) and then reared under laboratory conditions (16:8 L:D—temperature 24–25 °C—RH 60–70%). Fresh eggs are whitish; this color does not change until hatching, except for the appearance of a central black spot (black head stage) in the absence of parasitism. If the eggs are parasitized, their color changes from whitish to blackish; if only part of the egg mass is parasitized, both colors can be observed [1].

Egg masses kept under laboratory conditions were checked daily using a stereomicroscope to detect any color changes. Eggs were counted when embryo development was clearly defined. Finally, the parasitised/nonparasitised ratio was obtained by calculating the weighted average (inner and edge subsamples). Parasitism rate was calculated as the ratio between parasitized eggs and the total amount of eggs and expressed as a percentage.

Samplings of *T. brassicae* were carried out after treatments. In 2008, the insecticide “Daskor” (2 kg/ha) was sprayed on 29 July. In 2009, insecticides were sprayed twice: a first application of “Daskor” (2 kg/ha—29 July 2009) followed by a spray of alpha-cypermethrin on 11 August 2009 (“Contest”—0.5 kg/ha).

### 2.3. Statistical Analysis

Biostat 5.7.4 software (Analysoft) was used. Contingency tables were used to analyze the relationship between two or more categorical variables with the aim of identifying a possible interaction between them. They supported the analysis of data on plants infested by aphid colonies and aphid/predator rates. In addition, parasitism rates on ECB egg masses (calculated as follows: number of parasitized eggs/eggs total number×100) in sprayed and unsprayed plots were compared using contingency tables. 

## 3. Results

### 3.1. Impact of Insecticides on Aphids and Their Predators

#### 3.1.1. Aphid Community Composition

Maize fields were colonized by aphids in June–July; at this stage of plant development they were mainly distributed at the top of the plant, next to the male inflorescence. In August aphid colonies were mainly concentrated on the ears.

Table 1 summarizes the data on the frequency distribution of aphid species. The contingency table shows no significant difference between the Study Areas (*p* > 0.05). The dominant role of *Rhopalosiphum padi* (L.) is confirmed by data obtained as a result of a survey (2008–2009) in other maize fields located around Study Area 1 (*R. padi* = 64.8%—aphid colonies n = 55).

The number of maize plants hosting aphids in 2008 in the inner samples does not significantly exceed that observed in the edge samples. On the contrary, the data recorded in 2009 (Table 2) show a higher abundance of aphids in the edge samples compared to those in the inner part of the maize fields (contingency table—*p* < 0.01). No difference was observed in 2010 (contingency table—*p* > 0.05).

#### 3.1.2. Aphids on Maize and Their Predators (Years 2008–2010)

Figure 1 shows data on the percentage of maize plants hosting aphid colonies (year 2008). The acronyms Spr1 and Spr2 denote samples exposed to insecticide sprays in Study Area 1 and Study Area 2, respectively.

The comparison between sprayed (Spr) and unsprayed (Uspr) fields in both Study Areas showed a highly significant difference (Contingency tables—*p* < 0.001). The same result (*p* < 0.01) was obtained when Spr and Uspr data from the two Study Areas were compared (*p* < 0.01).

In summary, aphid colony density in unsprayed samples was significantly lower than in samples from sprayed fields and both Study Areas showed such a trend. Aphid density in Study Area 2 was significantly higher than in Study Area 1; this trend affected both Spr and Uspr samples. Regarding the natural enemies of aphids (Figure 2), the proportion of maize plants infested by aphids that host predators in the Uspr samples was significantly higher than that recorded in the Spr fields (contingency table), in both Study Area 1 (*p* < 0.01) and in Study Area 2 (*p* < 0.05). It should be noted that the natural enemy samples of Uspr1, Uspr2 and Spr1 were dominated by ladybirds (>50%), whereas the natural enemy community in Spr2 was dominated by hoverflies (97%).

In 2009, insecticide applications in Study Area 2 induced a significantly higher abundance of aphid colonies on sprayed maize (Figure 3) compared to the control sample (contingency table—*p* < 0.001). In Study Area 1, this trend was not recorded (*p* > 0.05). On the other hand, the frequency of predators on plants hosting aphid colonies was not significantly different in the two Study Areas (Figure 4—contingency table—*p* > 0.05).

In 2010, no insecticide was applied in both Study Areas. As a result, a sharp decrease in aphid abundance was recorded in S. Stefano (Study Area 2) compared to 2008 and 2009 (Table 3), when this area was sprayed (contingency table—*p* < 0.001), although the level of infestation remained higher than that recorded in the Study Area 1 sample (contingency table—*p* < 0.001).

#### 3.1.3. Predator Community: Composition and Response to Aphid Density

The predator community included ladybirds, hoverflies, lacewings (*Chrysoperla* sp.), bugs (*Orius* sp., *Nabis* sp.), earwigs (*Forficula* sp.), and beetles (Staphylinidae). The proportion of maize plants colonized by predators tended to increase with increasing aphid infestation according to a density-dependent pattern (contingency table—*p* < 0.01) (Figure 5).

Data on the composition of the predator community in the Study Areas are summarized in Table 4. Ladybirds and bugs (*Orius* sp.) were the most abundant predators in Study Area 1 from year to year. This was also the case in Study Area 2 in 2010, when no insecticide was applied. A significant dominance of hoverflies (97% and 56.9% in 2008 and 2009, respectively) was recorded in (Spr2), where a repeated maize crop pattern was practiced and insecticides were sprayed.

Adult ladybird populations (n = 191) included *Hippodamia variegata* Goeze (51.8%), *Harmonia axyridis* (Pallas) (16.8%), *Propylaea quatuordecimpunctata* (L.) (16.2%), *Scymnus* sp. (8.4%), and *Coccinella septempunctata* L. (6.8%). The sample of reared hoverfly pupae (n = 35) consisted of *Episyrphus balteatus* (De Geer) (80%) and *Sphaerophoria* sp. (20%). It should be noted that 80% of the hoverfly pupae were parasitized by *Pachyneuron* sp. (Hymenoptera: Pteromalidae) and *Diplazon laetatorius* (Fabricius) (Hymenoptera: Ichneumonidae).

### 3.2. Impact of Insecticides on Trichogramma brassicae

The typical development of second-generation ECB egg parasitism by *T. brassicae* is shown in Figure 6 (unsprayed sample): the parasitism rate was moderate at the beginning of oviposition by second-flight ECB females, but it tends to increase exponentially in August and usually peaks between the end of August and the first ten days of September, when parasitism rates can exceed 90% [19].

Figure 6 includes data from both unsprayed and sprayed samples recorded in 2008 and shows the parasitism development from the beginning of August after insecticide spraying. Parasitism in the control sample was significantly higher in the first two weeks of August (contingency table—*p* < 0.05). The effect of chemical spraying on sweet maize (2009 season) can also be deduced from Figure 7. In this case, it is important to note that the insecticide was sprayed twice.

## 4. Discussion

This case study investigated the impact of insecticide side effects on some natural enemies of maize pests. These effects depend mainly on the chemical nature of the active ingredients and the dose applied, and can cause lethal and/or sublethal effects [20,21,22,23,24].

The aphid community was dominated by *R. padi*. This species has a cosmopolitan distribution and is a common pest of various cereal crops [25]. A double effect caused by insecticide applications was evidenced; both aphid abundance and the predator community were affected. The abundance of aphid populations in maize was significantly lower in the Study Area 1 than in the Study Area 2, and such a discrepancy can reasonably be related to a different cropping pattern (Figure 1 and Figure 3). Unlike Study Area 1, where maize was part of a rotation protocol, Study Area 2 was managed according to repeated maize crops. The use of insecticides in combination with repeated maize crop may have the side effect of increasing aphids, which are usually secondary maize pests in chemical-free maize crops [26,27].

The aphid predator community in chemically sprayed samples in Study Area 2 in 2008 and 2009 was dominated by hoverflies, while ladybirds, *Orius* spp. and other natural enemies were rare or absent. Such a compositional spectrum differed significantly from that recorded in unsprayed fields. In addition, the aphid predator community (Table 4) changed profoundly and rapidly in 2010 when the maize field in Study Area 2 was no more subjected to insecticide spray: ladybirds and *Orius* spp. became dominant over other predators. Hoverflies are strong flyers; such an ability may suggest that they can colonize maize fields more quickly after insecticide spraying, when the presence of new aphid colonies after the insecticide treatment attracts egg-laying hoverflies. Predators, such as *Orius* spp. are, like ladybirds or hoverflies, common natural enemies of ECB in maize grown in the Po Valley. *Orius* spp. can be used as an indicator insect for evaluating the effect of insecticide treatments [28].

With respect to the genus *Trichogramma*, there is an extensive bibliography on the effects of insecticide applications, given the importance of this egg parasitoid as a potential biocontrol agent in many countries [20,21,22,23,24,29,30,31,32,33,34,35,36,37]. Several active ingredients have been tested under laboratory conditions: carbamates [30,32,35,36] organophosphorus compounds [20,21,24,33,38,39,40], pyrethroids [22,23,31,37,40], growth regulators [30,32,38,41] neonicotinoids [32], and spinosad [32,33,35,38]. Both sublethal and lethal effects have been observed. 

We investigated the impact of chlorpyrifos and cypermethrin: some data on effects coming from these active ingredients are still available. The negative side effects of sublethal doses of chlorpyrifos on *T. brassicae* have been documented; both sex pheromone communication and sex ratio were negatively affected [20,21,24]. Deltamethrin exposure may also alter pheromone-mediated sexual interactions [22,23]. Parsaeyan et al. [37] demonstrated the effects of cypermethrin on *T. brassicae* at recommended field concentrations, resulting in 80.7% mortality of pre-imaginal stages under laboratory conditions.

Our research has shown the negative effects of insecticide applications on *T. brassicae*. A change in the trend of natural parasitism was observed. Parasitism rates tend to increase and peak at the end of the season (September) in the absence of disturbance factors. In 2008, a negative effect occurred in the first fortnight of August, while no significant difference in parasitism rates was recorded later in the season (Figure 6). This trend may be related to the fact that *T. brassicae* can complete a generation in 10 days at 25 °C [42] and can move and parasitize ECB egg masses when insecticide residues are not harmful anymore. The effect of insecticide spraying was much more pronounced in 2009 due to the double chemical application (Figure 7). After an initial decrease in *T. brassicae* density caused by the first insecticide application, the recovery of parasitism rates was inhibited by the second insecticide application.

## 5. Conclusions

Selectivity is one of the desired properties of an insecticide, but often the target pest is not the only victim of spraying, and beneficial insects are usually affected to some extent [43,44]. Even if natural enemies can survive insecticide application, sub-lethal doses can cause chronic toxicity, affecting the behavior and fitness of organisms in subsequent generations [45]. For these reasons, the conservation strategy for beneficial insects in crops where insecticide use is part of the pest management protocol is a difficult one.

Farm management is critical in promoting or preventing pest outbreaks: for example, as this case study shows, repeated maize crop tends to encourage pest density, which in turn may require insecticide treatments [6,9]. Another consequence of chemical applications could be the increase of a pest other than the one targeted. Indeed, this case study reported an increase in aphid populations as a result of insecticide. Although monoculture and repeated maize cropping have been discouraged for decades by IPM extension service technicians, they are still fairly common practices in some areas of Po Valley and other European maize agroecosystem that tend to act as a predisposing factor for pest outbreaks. In general, insecticides use in maize cultivations are not economically consistent, due to the so-called pesticide treadmill outcome.

Biological control and integrated pest management can be a viable alternative to synthetic chemical sprays. Recent advances in research on the sustainable management of ECB demonstrate that a biological control strategy based on the inundative release of *T. brassicae* could be a viable approach to limit the second generation of ECB larvae without increasing mycotoxin levels in grain and yield losses [1]. Such an approach is consistent with the goal of reducing pesticide use, which is a core strategy of public policy aimed at restoring the environmental quality of agricultural lands [9,46].

## Figures and Tables

**Figure 1 insects-15-00224-f001:**
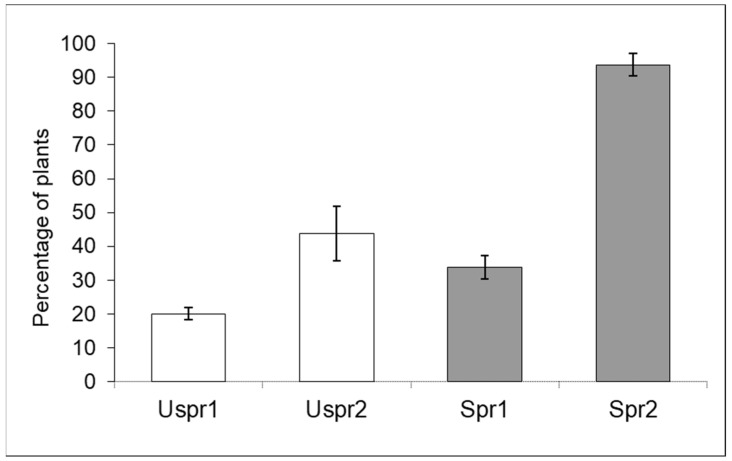
Aphid-infested maize plants (number of inspected plants/sample = 160) in sprayed (Spr) and control/unsprayed (Uspr) fields in 2008 (% ±S.E.). Study Area is indicated by 1 or 2.

**Figure 2 insects-15-00224-f002:**
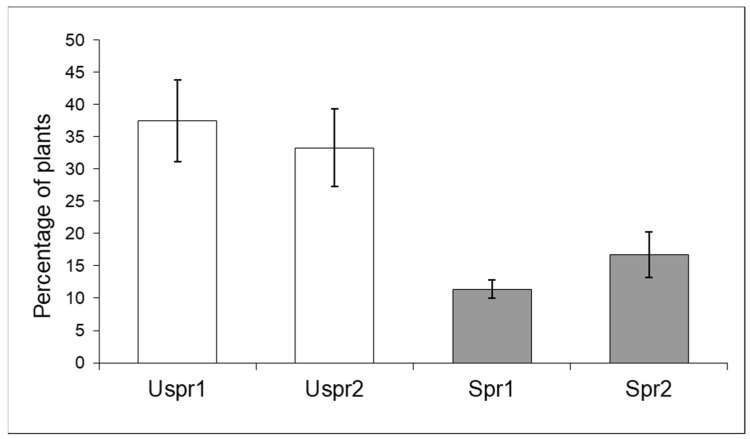
Maize plants infested by aphids that host natural enemies (% ±S.E.) in sprayed (Spr) and control/unsprayed (Uspr) samples in 2008. Study Area is indicated by 1 or 2.

**Figure 3 insects-15-00224-f003:**
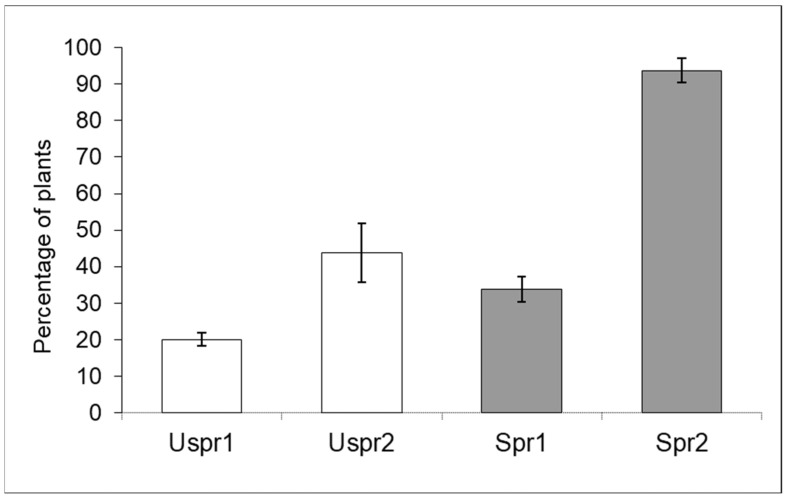
Aphid infested plants (% ±S.E.) in sprayed (Spr) and control/unsprayed (Uspr) samples in 2009. Study Area is indicated by 1 or 2.

**Figure 4 insects-15-00224-f004:**
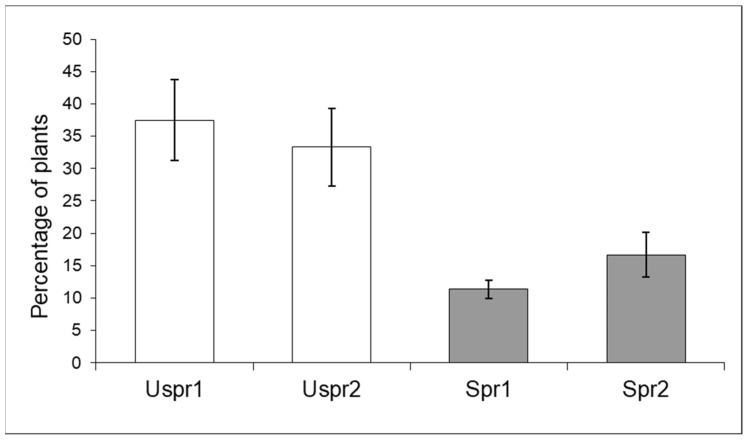
Maize plants infested by aphids hosting natural enemies (% ±S.E.) in sprayed (Spr) and control/unsprayed (Uspr) samples in 2009. Study Area is indicated by 1 or 2.

**Figure 5 insects-15-00224-f005:**
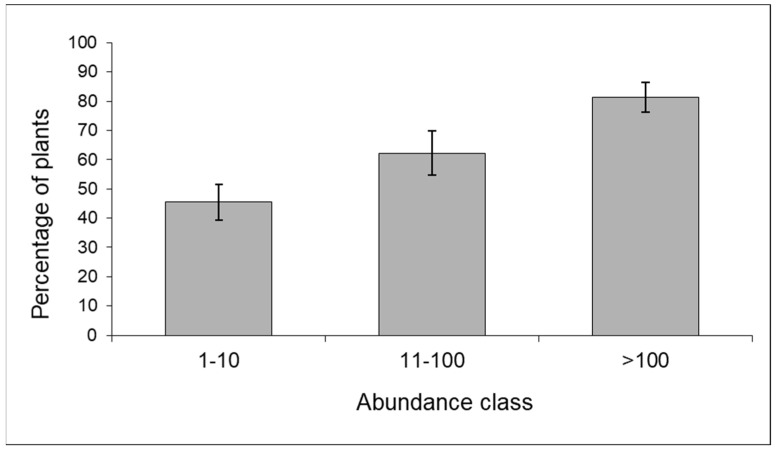
Frequency distribution of maize plants colonized by predators in relation to aphid abundance; data from 2009 (n = 241) and 2010 (n = 195) (% ±S.E.).

**Figure 6 insects-15-00224-f006:**
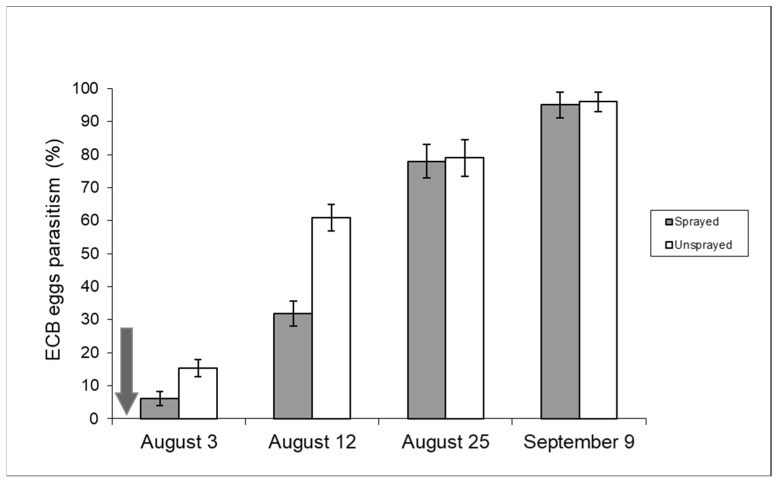
Parasitism (%) of ECB eggs recorded in 2008 in unsprayed and sprayed maize fields (year 2008). The arrow points to the day (29 July) of insecticide application [Uspr: egg masses n = 88, eggs/egg mass 33.8 ± 0.98 (±S.E.); Spr: egg masses n = 91, eggs/egg mass 34 ± 0.68 (±S.E.)].

**Figure 7 insects-15-00224-f007:**
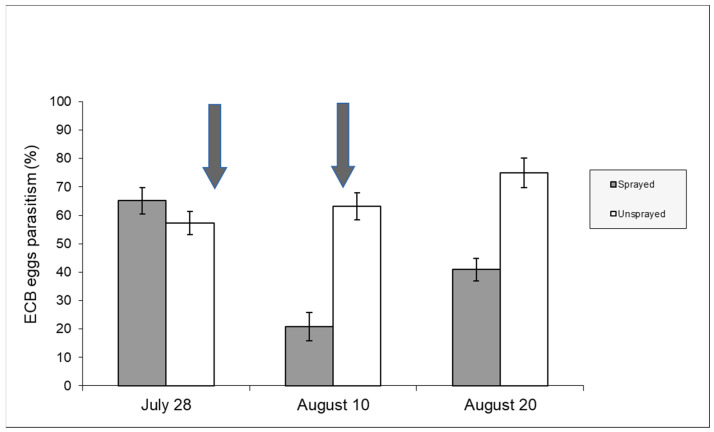
Parasitism rates (%) recorded in sweet maize (year 2009). The arrows indicate the days (29 July—11 August) when the insecticide was applied [Uspr: egg masses n = 48, eggs/egg mass 33.83 ±1.26 (±S.E.); Spr: egg masses n = 41, eggs/egg mass 32.56 ± 1.42 (±S.E.)].

**Table 1 insects-15-00224-t001:** Frequency distribution of aphid species from samples collected in Study Area 1 (n = 46) and Study Area 2 (n = 35)—(years 2008–2010; weighted average; samples from both sprayed and unsprayed fields).

Species	Area 1	Area 2
*Rhopalosiphum padi* (L.)	64.8	59.9
*Rhopalosiphum maidis* (Fitch)	20.8	25.6
*Sipha maidis* Passerini	10.4	11.3
*Sitobion avenae* (F.)	4	3.2
*Aphis* sp.	0	0

**Table 2 insects-15-00224-t002:** Frequency distribution (%) of aphid abundance in the edge and inner samples in 2009 and 2010 (samples from both sprayed and unsprayed fields).

Sample	Year	1–10	11–100	>100	n
Edge	2009	30.1	24.8	45.1	133
Inner	2009	27.8	42.6	29.6	108
Edge	2010	64.6	30.2	5.2	96
Inner	2010	71	20.4	8.6	89

**Table 3 insects-15-00224-t003:** Percentage of infested plants and aphid-hosting plants with predators (year 2010). Each sample included n = 160 maize plants (±S.E.).

Sample	Plants with Aphids (%)	Plants with Predators (%)
Area 1	48.8 ± 6	97.1 ± 1.9
Area 2 (Robecco S.N.)	15.6 ± 1.2	61.7 ± 7.8
Area 2 (S. Stefano T.)	57.5 ± 2.5	44 ± 4.9

**Table 4 insects-15-00224-t004:** Predator frequency (%) in Study Areas (year 2008—n = 96; year 2009—n = 200; year 2010—n = 185). Data from 2008 and 2009 include sprayed and unsprayed samples. No insecticide was sprayed in 2010 in either Study Area.

Predators	Year 2008		Year 2009		Year 2010	
	Area 1	Area 2	Area 1	Area 2	Area 1	Area 2
Ladybirds	84.3	26.4	38.4	26.4	78.3	26.7
*Orius*	9.1	18.5	37.8	19.2	13	63.9
Hoverflies	6.2	49.9	9.7	33.4	1.7	7.5
Lacewings	0.3	0.2	9.4	17.7	4.4	1.9
Others	0.1	5	4.7	3.3	2.6	0

## Data Availability

The original contributions presented in the study are included in the article material; further inquiries can be directed to the corresponding author.

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
