# Peer review of "Impact of Chemical Insecticide Application on Beneficial Insects in Maize"

_insects, 2024, doi:10.3390/insects15040224_

Round 1

Reviewer 1 Report

Comments and Suggestions for Authors

The impact of insecticide applications on pest insects and natural enemies in agriculture are an important and often discussed field of research. Agricultural practices (such as crop rotation) help reduce pest populations and increase beneficial insects. In addition with reduced or no pesticide applications, the positive effects may even be amplified.  

While your research addresses this important topic, this manuscript does not adequately represent the outcome of your studies. Your introduction needs a common thread as well as a more precise focus on your research questions. Why include all this information on WCR when this pest is not even part of your studies? Your material and method section requires additional details on how you conducted and analyzed your research. Your discussion in general is weak and unfortunately rather superficial. It needs to be more focused on putting your results in the context of published literature in this field of research. 

Introduction: 

Introduction needs to be more precise - the switch back and forth between ECB and WCR is confusing and the common thread as a lead-in to your research is weak. The literature cited in the introduction is limited to a few publications {1-4} but there is a vast amount of international publications on these two pests, e.g. on their biology as well as the biological control component, that would be appropriate to cite in this context. 

Why did you include all this information on WCR when no part of your research was aimed towards this pest? 

Materials and methods:

2.1: 

- Why did you not also monitor diel flight activity in area 2? 

- Your study areas differ by the type of maize grown - feed and sweet corn versus maize for silage. Did you account for this variable in your analyses? 

- What were the dimensions of the fields use? Were field sizes and plot sizes comparable between area 1 and 2?

- what was the weather like in the two study areas? Were the conditions comparable?

- Pest populations can significantly differ between years, even at the same locations, e.g. due to the amount of rain or heat. Please include information on temperature, precipitation etc. at your locations during the time of your studies.

2.2.1: 

- Parasitoids are important natural enemies of aphids and commonly found in aphid-infested crops. Why did you not include them in your monitoring? Collecting mummies, e.g., as you did for hoverfly pupae, could have been easily incorporated in your regular sampling schedule.

  • Please clarify during which time periods samples were taken in each year and how often samples were taken. 
  • As for the predators you monitored, what developmental stages did you record for for each species in general and for hoverflies (besides pupae) in particular? Only Syrphid larvae exert biological control on aphids. 

2.2.2: 

lines 158-161: Is this the same spray regime as already outlined in lines 135-139 or were these insecticides applications in addition to them? Why mention them twice?

2.3: 

What statistical program(s) did you use? How did you calculate parasitism rates? What were your variables? Please specify. 

Results: 

- line 191: What do the numbers represent? Weighted averages as in Table 1? Why are data for 2008 not shown as well? Are data pooled for area 1 and 2? Specify. 

- Are the data shown here pooled for inner and edge samples? 

  • Figures 3 and 4: data shown for 2008 or 2009? Text and figure text not identical. 
  • Figures 1-4: Specify on the y-axes themselves what exactly is shown on the y-axis!
  • Does figure 6 show parasitism of individual eggs or egg masses? Clarify.

Discussion: 

lines 278-280: while this statement is true, none of the results shown supports

  • This line of reasoning is unclear as you do not clearly discriminate between the effects of crop management (crop rotation vs. monoculture) and insecticide applications. In addition, you are comparing two separate areas as well as different crop types (sweet corn, field corn, maize for silage).  There may have also been differences between the field locations, e.g. abiotic and biotic factors specific to each location, that may have had an influence on your pest populations. Please clarify if you accounted for these variables in your statistical analyses. 
  • In study area 1, there were also insecticides applied during your study, Thus, your reasoning (esp. lines 287-289) is unclear to me.  
  • line 320-321: and yet in comparison, parasitism rates in sprayed fields increased after the second application compared to the first one. How do you explain this? 

General remarks:

line 15: egg parasitoid

line 23: include the scientific name for ECB here

line 33: bugs is the slang term for insects. Do you mean true bugs (Hemiptera)?

line 36: just threshold or alternatively use the term economic threshold

line 61: attacks on ear shanks

line 62-65: Was the observed flight in mid-September a one-time occurrence or has it been observed on a more regular basis? Please clarify and, if needed, adjust the grammar of your sentence. 

line 71-72: this is phrased in an awkward and somewhat misleading way as Trichogramma sp. is active when egg masses of the host are present. Consider re-phrasing. 

line 267: evolution is not the appropriate term in this context. Use e.g. development over the years. 

Comments on the Quality of English Language

The quality of english used throughout the manuscript needs some improvement, i.e. correct use of scientific terms, english grammar (plural versus singular, cases). 

Author Response

Thank you for your revision, we implemented the paper according to your suggestions. Here attached you can find our answer.

Reviewer 2 Report

Comments and Suggestions for Authors
    • GENERAL COMMENT:
      •  

    The work presented shows potential, aiming to examine the effects of certain insecticides on predators and parasitoids of maize phytophages. However, it is essential to subject it to a thorough review, particularly concerning the description of the methodologies used. It is also advisable to highlight how some of the insecticides analyzed in this study in the years 2008, 2009, and 2010 are no longer available on the European Union market today, in 2024. Such observation could significantly enrich the conclusions of the study. Moreover, the introduction needs to be expanded and refined. Additionally, there is no mention of B. thuringensis in the abstract.

    There is also a lack of attention in the use of symbols, such as "%" in tables, and in the use of spaces (for example, one should write "P > 0.05" instead of "P>0.05"). Regarding Table 2, although the description refers to 2009, data from 2010 are also present, suggesting the need for a correction to ensure consistency and clarity.

    SOME SPECIFIC COMMENTS

    • Line 23: Remove the first comma after "ECB" and change "and in a lesser extent" to "and to a lesser extent".
    •  
    • Line 27: Change "A three-year research" with "A three-year research project"
    •  
    • Lines 35-38 Suggestion for clarity: The phrase "when the threshold limit can be at risk" could be better clarified. It might be better to specify what threshold is being referred. (e.g. "Crop rotation protocols help prevent ECB outbreaks. When the risk of surpassing the threshold limit is high, the inundative release of T. brassicae or microbial control, such as the use of Bacillus thuringiensis preparations, can supplement natural biocontrol methods, providing a valuable alternative to chemical insecticides.")
  1.  
  2. Lines 43-44 some references should be added.
  3.  
  4. Line 46 change entomological threat with pest and add "," after More recently
  5.  
  6. Line 47 change such as with like and change by the new with by a new

     Line 53 change ":" with ","

     Line 55-64 i suggest rephrasing this paragraph.

Other General comments:

Materials and Methods:

Study areas

I have not understood how many fields were investigated in Area 1.

I believe that most of the information presented from line 85 to line 108 can be summarized in a table, which would make it easier to memorize the information

Aphids

I believe that including a figure could make it easier to understand how and where samplings were performed.

Have you not sampled the population of aphids and predators before the application of insecticides?

Statistical Analysis

The description of statistical methods needs to be improved. Which test did you use?

Results

Table 1 should indicate 64.8, not 63.8.

Discussion

Change the period to a comma at line 307.

  1.  
  2.  

Author Response

Thank you for your revision. In the attached file our answer.

Round 2

Reviewer 1 Report

Comments and Suggestions for Authors

Dear authors,

thank you for your revision and for addressing the comments and suggestions on your manuscript. 

Comments on the Quality of English Language

There are still some minor improvements to be made throughout the document - e.g. incorrect use of singular/plural. 

Reviewer 2 Report

Comments and Suggestions for Authors

I am thankful for the changes that have been made, I have nothing to add.